# Photochemical Metallization: Advancements in Polypropylene Surface Treatment

**DOI:** 10.3390/polym15183687

**Published:** 2023-09-07

**Authors:** Bagdagul Serikbayeva, Malik Satayev, Shaizada Koshkarbayeva, Abdugani Azimov, Kalamkas Amanbayeva, Guzaliya Sagitova, Aliya Suigenbayeva, Myrzabai Narmanov, Artem Kolesnikov

**Affiliations:** 1High School of Chemical Engineering and Biotechnology, M. Auezov South Kazakhstan University, Shymkent 160012, Kazakhstan; malik_1943@mail.ru (M.S.); shayzada-1968@mail.ru (S.K.); azimov-78@mail.ru (A.A.); kalam.70@mail.ru (K.A.); guzalita.f1978@mail.ru (G.S.); syaljo@mail.ru (A.S.); narmanov@bk.ru (M.N.); 2Faculty of Technology of Inorganic Substances and High Temperature Materials, Mendeleev University of Chemical Technology of Russia, 125047 Moscow, Russia; artkoles@list.ru

**Keywords:** polymer, metallization, silver, ascorbic acid, photochemical processes

## Abstract

The work was devoted to the development of technology for applying metal coatings to the surface of polypropylene products. At the same time, the main stages of the technology were carried out using the influence of electromagnetic waves of light radiation. So, to obtain an electrically conductive silver layer, after mechanical treatment, etching and activation, the polymer was immersed for several minutes in a solution containing 10–20 g/L of silver nitrate and equivalent amounts of ascorbic acid, and a thin layer of solution was obtained on the surface of the polymer. A sample with such a sorption film was exposed to electromagnetic waves of light radiation at a flux density of 700–1100 W/m^2^. The small thickness of the sorption film facilitated the penetration of these waves directly onto the polymer surface and ensured the photochemical process of silver reduction with the formation of active centers. At the same time, electromagnetic waves acting on ascorbic acid transferred it to an excited state. As a result, the chemical reduction of silver in the space between the active centers became possible. In this case, the film obtained within 15–20 min had the necessary electrical conductivity. The suitability of these films for galvanic metallization of the polymer surface was shown.

## 1. Introduction

Metal coatings are applied to the surfaces of polymeric materials to impart new functional properties to them: electrical conductivity, hardness, wear resistance, etc. The methods known in the literature for depositing metal films on dielectric materials are conventionally divided into physical and chemical ones.

Of the physical methods, thermal spraying is most often used. In this process, pre-melted metal particles are sprayed onto the surface to be coated with compressed air or gas. Particles of liquid metal on the way to the surface of the dielectric are partially cooled and only melt the surface, sticking to it [1,2,3,4,5]. 

Technological progress has contributed to the emergence of new varieties of this method. These are gas-phase, plasma and vacuum deposition [4,6]. Vacuum metallization technology is distinguished by its versatility, environmental friendliness and operating personnel. Modern technologies also include magnetron sputtering in a vacuum chamber of metal particles (including silver) with subsequent deposition on the surface of the material. The method is based on the use of an anomalous glow discharge in an inert gas, in which positively charged ions formed in the discharge bombard the cathode surface in the erosion zone and knock out metal particles from it, which are then deposited in the form of a thin layer on the surface of the material being processed [7,8,9]. In this case, the high kinetic energy of the particles leaving the cathode surface ensures a good level of adhesion of the resulting film to the substrate. 

It is effective to use metal carbonyls Me_m_(CO)_n_ in these methods, which are complexes of transition metals with carbon monoxide. The thermal decomposition of cobalt, nickel, and chromium carbonyls is used for the deposition of metal coatings, especially on dielectric surfaces. The carbon monoxide released in this case can again be used to obtain a metal carbonyl. That is, CO plays the role of a metal carrier reagent. This is not only convenient in terms of production but also minimizes unproductive costs of auxiliary reagents and eliminates environmental pollution. The widespread use of this method is hindered by the relatively high temperature of the process and the high cost of metal carbonyls [10]. 

The main disadvantage of physical methods is the need to use complex expensive equipment. Therefore, the use of this method is economically justified in the metallization of the same type of dielectric products in significant quantities (electronics, vehicles, etc.).

Chemical methods of metallization of dielectrics consist of carrying out chemical processes for obtaining metal particles in a gas or liquid medium. 

In the study by Koshkarbayeva et al. [11,12], they proposed to initially obtain copper phosphide films on fabric materials moistened with copper sulfate by exposure to gaseous phosphine PH_3_, which were then transformed into silver. The possibility of using this method for both synthetic and natural tissues was shown. The disadvantage of the method was the use of toxic phosphine as a reducing agent and the resulting need to work in sealed conditions.

As the review by Ghosh [13] showed, the metallization of dielectrics in the liquid phase is the most common technology, which contains the operations of sensitization and activation (separately or combined) and almost always involves the use of precious metals (mainly palladium). As a result of activation, nuclei are formed on the surface of the dielectric, which serve as a catalyst for the subsequent chemical metallization reaction and results in the formation of a conductive sublayer. In the chemical–electrolytic method most commonly used in technology, the conductive layer is created from copper. Therefore, the main stage of such metallization is the process of chemical copper plating. On the electrically conductive layer thus obtained, a thicker layer of copper is usually applied by electrochemical deposition and then any other metal that performs the functions of the main metal coating.

Kapitsa [14,15] showed the results of works on chemical copper plating of the surface of printed circuit boards using formaldehyde as the reducing agent. It has been noted that such a process ensures the good quality of printed circuit boards. However, Ding J. and Retallik R. in [16] noted the presence of environmental problems in the treatment of wastewater generated after chemical copper plating.

Technological solutions of chemical copper plating containing copper salts, complexing agents and formaldehyde are highly toxic. Wastewater treatment is complicated by the presence of chelate compounds, which are formed from copper ions and organic residues of complexing compounds [16]. These compounds prevent the release of metals from wastewater, which makes the process of their treatment very laborious and, therefore, expensive. 

In this regard, alternative technologies have been created that exclude the stage of chemical copper plating from the metallization process. This has also been facilitated by legislative acts on environmental protection adopted in most states.

In [17], Lundquis, J. proposed a direct metallization process as such an alternative technology, the peculiarity of which was that, instead of individual catalytically active inclusions (catalyst), a conductive palladium film was created on the surface of the dielectric, on which copper could immediately be deposited by using the galvanic method. Shkundina S. [18] presented the results of the introduction of the direct metallization process by J-KEM International AB (Sweden), in which a very thin palladium film made it possible to exclude chemical copper plating in the production of printed circuit boards. The experience of introducing and mastering the process of direct metallization of printed circuit boards Neopact by Atotech during production was shown in the work of Serzhantov A. [19]. It has been noted that the transition from conventional chemical copper plating using palladium particles as a catalyst to direct plating with a palladium film in the production of printed circuit boards has proven to be beneficial, since the number of operations has decreased and the higher stability of the chemical plating line.

In the work of Kapitsa [20], it was proposed to combine direct activation with etching in a chromic acid solution of the following composition to obtain ultrathin palladium films: CrO_3_—350 g/L; PdCl_2_—0.5 g/L; H_2_SO_4_—30%. This contributed to the simplification of technology by reducing the number of common operations.

For direct metallization of the surface of dielectrics, other metals with a high reduction ability can also be used. Thus, in a US Patent [21], the deposition of metallic silver on the surface of a dielectric material was studied using aqueous solutions containing a water-soluble silver salt (0.1–20 g/L), an ammonium hydroxide complexing agent, an ammonium carbonate stabilizer and a reducing agent hydrazine hydrate (N_2_H_4_·H_2_O) in quantities of 0.1–10 g/L. The disadvantage of this method is the low rate of metallization (at 60 °C and a duration of 20 min, 0.2–0.4 microns are deposited).

In a German Patent [22], to obtain thin electrically conductive silver layers on the substrate surface, at the first stage of the process, silver nitrate and 2-pyrrolidone are dissolved in water or an ethanol–water mixture to form [Ag Pyl)_2_]NO_3_. After that, in the second stage of the process, the solution is applied to the surface of the substrate to be coated and then by irradiation with electromagnetic waves of the UV spectrum for at least 15 min. In this case, a chemical reduction process occurs, which leads to the release of silver, followed by a heat treatment stage at a temperature of 220 °C for at least 60 min. 

The disadvantage of this method is the multistage nature and the need for additional equipment for heat treatment and irradiation with electromagnetic waves of the UV spectrum. Also, high temperatures make this method unsuitable for some polymeric materials. 

This review shows the relevance of direct metallization of the surfaces of dielectrics. In the proposed work, the authors investigated the process of low-temperature direct deposition of a thin electrically conductive silver film on polymeric materials using a sorption layer of solutions obtained by wetting the product in appropriate solutions.

## 2. Materials and Methods

Polymer plates made of PP H030 polypropylene were used for the research. In order to provide visual observation of the course of chemical and photochemical processes, samples were selected from polymers with light tones.

Preliminary preparation of the plates consisted of surface treatment with sandpaper P-2000 etching for 5–10 min at room temperature in a solution of K_2_Cr_2_O_7_—6.5% and H_2_SO_4_—93.5% and degreasing in a solution of Na_3_PO_4_—20 g/L and Na_2_CO_3_—20 g/L.

The processes of activation and deposition of silver films were carried out in a sorption film of solutions obtained by wetting the polymer surface by dipping into the studied solutions containing water-soluble salts of copper, silver, ascorbic acid and 1–3 g/L gelatin. All reagents used in the work were qualified as “chemically pure” and were not subjected to additional purification. At the same time, the thickness of the sorption film was about 400–450 microns. To determine the thickness of the sorption film, a plate with a size of 10 × 10 cm was cut out of the polymer, and this plate was lowered into the vessel with the test solution; then the sample was taken out and slightly shaken over the vessel. At the same time, the volume of the solution decreased. In order for the reduction to be measurable, these operations were repeated more than 10 times. The one-time carryover of the solution determined in this way by this plate was 8–9 cm^3^. The division of this value by the area of the plate, 10∙10∙2 = 200 cm^2^, showed us the thickness of the sorption film at 0.04–0.045 cm or 400–450 µm. This film contained all the compounds necessary for the process under study in the required quantities. For example, at a concentration of silver nitrate in the sorption layer of 20–40 g/L, it provided the formation of an electrically conductive silver film with a thickness of 0.1–0.48 microns on the polymer surface. At the same time, in order to prevent a decrease in the concentration of the reducing agent in the final period of the photochemical process in the sorption layer, a molar ratio of ascorbic acid with respect to silver nitrate of 2:1 was used. Such a sorption film allowed processes to be carried out by exposure to electromagnetic waves of light radiation (EMWLR). 

Solar radiation and radiation from artificial lighting lamps were used as a source of electromagnetic waves. The radiation flux density was determined by the SM 206-SOLAR solar radiation meter and was 800–1100 W/m^2^.

Ascorbic acid was used as the reducing agent. Since the formation of an electrically conductive silver film occurs when a certain concentration of ascorbic acid is reached, the dependence of the electrochemical potential on its concentration was studied. In these experiments, a P-4-type potentiostat (Russia) was used to determine the electrode potentials of an electrochemical cell in the process of electrochemical research. At the same time, an ascorbic acid solution was poured into the cell, an inert platinum electrode and a silver chloride reference electrode were immersed and the potential was measured.

The structure and composition of the films were studied using an ISM-6490-LV scanning electron microscope (JEOL, Japan). It is a scanning electron microscope (SEM), which creates images of a sample by scanning the surface with a focused electron beam. The electrons interact with the atoms in the sample, producing various signals that contain information about the surface topography and composition of the sample.

The elemental composition of the conductive films was determined using an EDX-7000 instrument (Shimadzu Corporation, Tokyo, Japan). The operation of the device is based on the X-ray radiation of the surface of the object and obtaining an image of the particles that make up the object. Then, using geometric optics, these mappings are obtained in the form of peaks characterizing a particular element.

In addition, to visually study the microstructure of the silver film formed on the polymer surface, this film was imaged using an ICX41M optical microscope (Sunnyoptical, SOPTOP, Ningbo, China) at 500-times magnification.

When determining the thickness of the silver film, the weight method used in electroplating was used. To do this, for analytical balance, the weight of the sample was low before and after the experiment. The resulting weight gain was divided by the product of the area of the sample and the density of silver.

The presence of electronic conductivity of the films was checked using a DT-830B digital multimeter, a universal device that allows one to measure the resistance, voltage and current. To determine the conductivity, the device was tuned to the appropriate resistance scale (ohmmeter mode). The multimeter’s probes then made contact with various parts of the film, providing a stable and strong connection.

The resulting conductive films were tested for the processes of galvanic copper plating and nickel plating, as well as the process of chemical nickel plating [23]. The adhesion of electroplated coatings to the polymer base was checked by the test proposed in [24].

## 3. Results and Discussion

As a result of preliminary preparation (mechanical and etching), a rough layer is formed on the surface of polymers containing hydroxyl and carbonyl groups belonging to aldehydes, ketones and acids, as well as double bonds [25].

When copper dichloride is introduced into the sorption layer, its interaction with the aldehyde group of the polymer R_n_CHO becomes possible by the reaction.

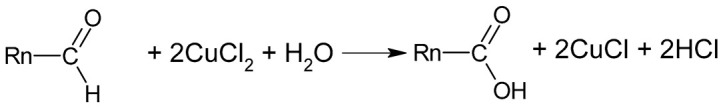
(1)

Since the aldehyde group of the polymer R_n_CHO is the solid-phase-associated part of the polymer, the resulting CuCl forms a strong bond with the polymer surface.

After washing the sample, active centers consisting of copper monochloride remain on the surface.

Then, the sample is moistened with a solution containing AgNO_3_ and ascorbic acid (C_6_H_8_O_6_) and exposed to EMWLR. 

At the same time, CuCl as a binary semiconductor under the influence of photons of light radiation is restored.
(2)$$CuCl\overset{hv}{\to }Cu+{Cl}^{-}$$

The electron hole (+) that appears in this case leads to the oxidation of ascorbic acid to dehydroascorbic acid. The equilibrium state is shown in Reaction 3.

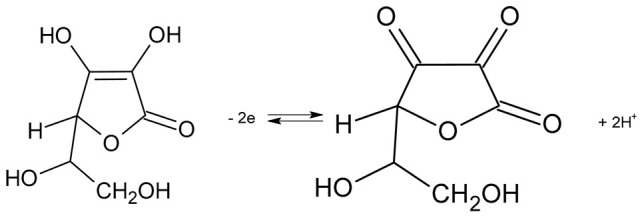
(3)

With this in mind, the total reaction occurring under the influence of photons of light radiation (photochemical reaction) will have this form:
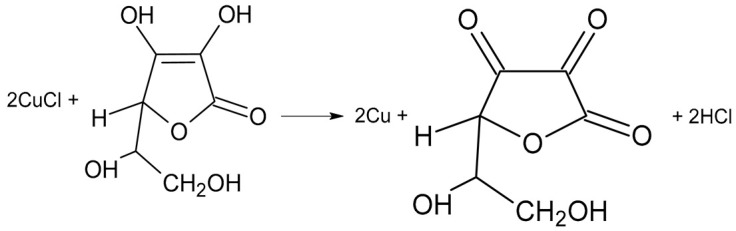
(4)

In this case, it is possible to replace CuCl with AgCl. This is facilitated by the fact that the solubility product of AgCl is several orders of magnitude lower than that of CuCl (K_SP(CuCl)_ = 1.2 × 10^−6^, K_SP(AgCl)_ = 1.78 × 10^−10^).

Silver chloride is a binary photochemically sensitive semiconductor; therefore, the following processes occur in the reaction medium containing ascorbic acid.
(5)$$AgCl\overset{hv}{\to }Ag+{Cl}^{-}$$

As a result of these photochemical reactions, silver ions are reduced to metallic silver.

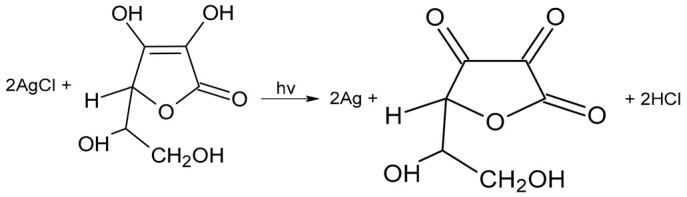
(6)

The resulting HCl interacting with silver nitrate contributes to the renewal of the photochemically active chloride. Therefore, the final photochemical reaction in the sorption layer will have this form:
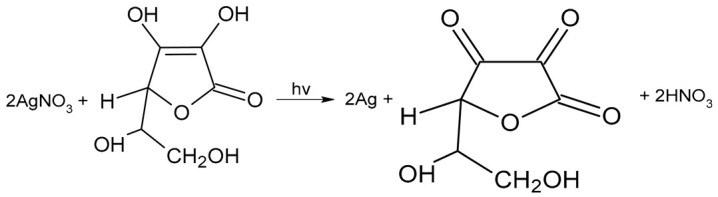
(7)

Since electromagnetic waves of light move at high speed strictly in a straight line, when passing through various bodies, there will always be sections (corridors) not covered by them. Such shadow areas show the resulting photochemical films have a black (or dark) color. In addition, these areas create barriers to the passage of electrons, and as a result, photochemical films formed by metal atoms are usually nonconductive.

In this case, the chemical reduction of silver with molecules of ascorbic acid is also possible according to the same reaction.

The chemical mechanism of the reaction is due to the fact that the redox potential of the ascorbic acid–dehydroascorbic acid system with a change in pH in the acidic region ranges from −0.329 V at pH 1 to −0.057 V at pH 7 [26]. This shows that the reduction of silver ions (E^o^ = +0.799 V) by ascorbic acid is a thermodynamically possible process.

Studies of the reducing properties by measuring its potential at various concentrations have shown that an increase in the reducing properties of AC is observed only before the concentration rises to 40 g/L (Figure 1).

At the same time, it has been noted in the literature [26] that the oxidation reaction of ascorbic acid under the action of silver nitrate is faster in light, but it can also be achieved the dark. It follows from this that this process, although chemical, requires the activation of ascorbic acid particles’ EMWLR. 

Conventional chemical reactions, unlike photochemical ones, proceed at the same speed in all directions. Therefore, silver reduction can also occur in voids formed after the photochemical process. Consequently, a combination of these two processes is required to obtain an electrically conductive silver sublayer. The use of ascorbic acid as a reducing agent makes it possible.

Table 1 shows the results of obtaining an electrically conductive layer by the interaction of silver nitrate with ascorbic acid under conditions of different illumination of the reaction medium. For example, in the dark, the reaction proceeds only by a chemical mechanism. The reaction speed is very small. There is a dependence on the concentration of reacting substances at their molar ratio of 1:2 (lines 1 and 5).

However, the influence of illumination is more noticeable. Even scattered daylight increases the speed of the process by almost an order of magnitude (lines 1, 2, 5 and 6). An even stronger influence on the reaction rate is exerted by direct light rays from the sun or an electric lighting device. At the same time, the reaction rate grows by several orders of magnitude (lines 1, 3, 4, 5, 6 and 7).

From the data in Table 1, it can be seen that the most effective process is to obtain an electrically conductive silver layer by exposing the EMWLR to a sorption film of a solution containing silver nitrate and ascorbic acid. At the same time, the volume of this film depends on the surface roughness, its wetting ability and the compositions of the reacting substances (Table 2).

The data in Table 2 show that, at low concentrations of silver nitrate (lines 1 and 2), the resulting film cannot completely cover the polymer surface with a solid and provide the desired electrical conductivity, and only at concentrations of more than 10 g/L, a continuous conductive silver film forms on the polymer surface. The thickness of this film depends on the concentration of silver nitrate in the sorption layer and is 0.11–0.48 microns (lines 3–6). The formation of such a solid electrically conductive silver film can also be seen in the SEM images (Figure 2).

The formation of a thin silver film is also confirmed by the data of X-ray phase analysis of the sample surface shown in Figure 3 and Table 3, where, in addition to the peaks of the elements that make up the initial polymer, metallic silver peaks appear.

The experiments make it possible to establish that, when the content of silver nitrate in the sorption layer on the polymer surface is 10–20 g/L and equivalent amounts of ascorbic acid are 10.05–20.1 g/L, silver films with electronic conductivity are formed when exposed to EMWLR. 

Diagrams of the processes occurring in this case are shown in Figure 4.

Polymer plates with an electrically conductive silver layer were tested in the processes of electroplating thick layers of copper nickel [24].

Galvanic copper plating was carried out in a sulfuric acid electrolyte of the composition (g/L):

CuSO_4_·7H_2_O—50;

H_2_SO_4_—50;

C_2_H_5_OH—50;

Nickel plating was carried out in an electrolyte of the composition (g/L):

NiSO_4_·10H_2_O—250;

NiCl_2_·6H_2_O—50;

H_3_BO_3_—30;

Succinic acid—20.

Photos of the samples obtained in this case are shown in Figure 5.

An analysis of the surface images (Figure 6) of these samples obtained using an electron microscope shows that the silver film has a pronounced crystalline structure. Silver crystals are larger than crystals of copper and nickel deposited and electroplated. In this case, the silver film is continuous and does not have areas with an amorphous polymer structure.

The adhesion of metal coatings to the polymer base was tested using adhesive tape [25]. In this case, no peeling was observed, which indicates satisfactory adhesion.

## 4. Conclusions

The use of a reaction medium in the form of thin sorption layers of a solution of silver nitrate and ascorbic acid in the technology of the metallization of polymers makes it possible to carry out the process under the influence of electromagnetic waves of light radiation. In this case, crystallization centers are formed due to the reduction of silver ions by photons of light radiation. At the same time, such radiation activates ascorbic acid molecules, which facilitates the reduction of silver ions in the spaces between crystallization centers. When these processes occur together, the necessary conditions are established for creating an electrically conductive silver film. Such a film allows the use of standard galvanic processes for the metallization of polymers with other metals.

## Figures and Tables

**Figure 1 polymers-15-03687-f001:**
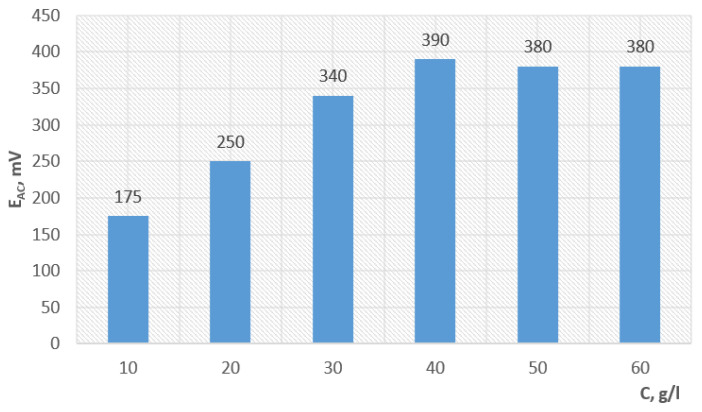
Dependence of E_AC_ of ascorbic acid on the concentration.

**Figure 2 polymers-15-03687-f002:**
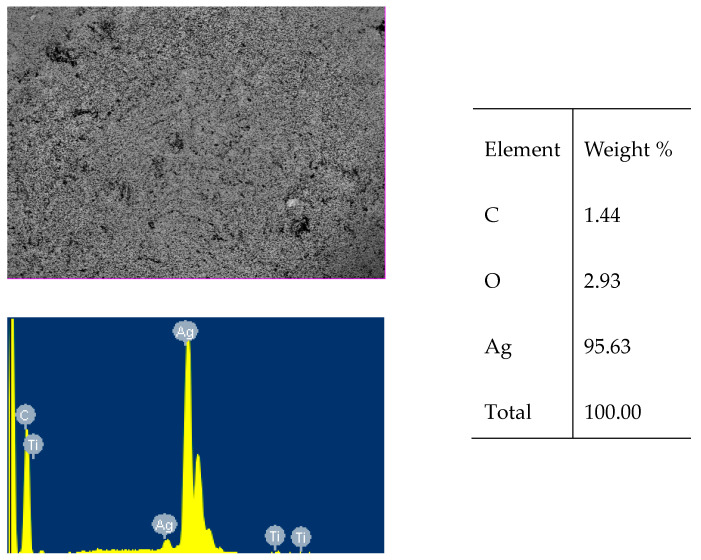
SEM image and elemental composition of a silver film obtained by exposure to EMWLR at a flux density of 80 W/m^2^ on a sorption layer containing 10 g/L of silver nitrate and 10.05 g/L of ascorbic acid.

**Figure 3 polymers-15-03687-f003:**
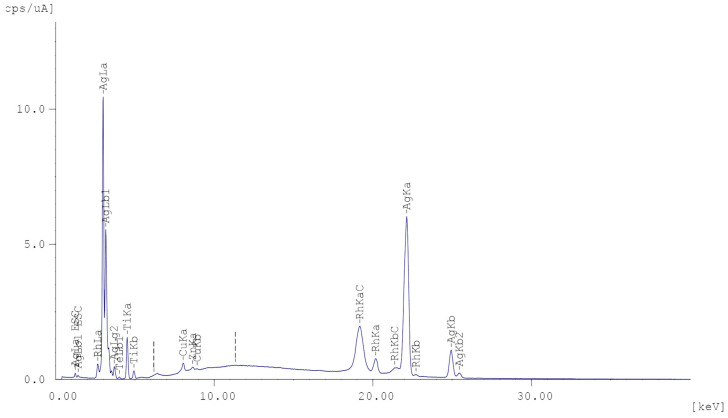
X-ray phase analysis of the sample surface.

**Figure 4 polymers-15-03687-f004:**
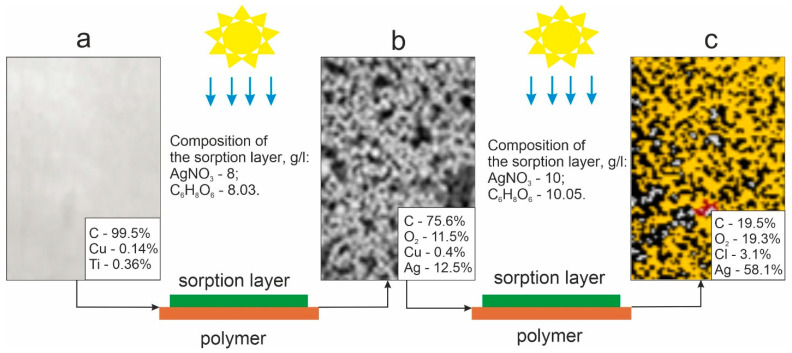
Processes in the formation of silver-containing films on the surface of polymers: (**a**) the surface of the plate after activation; (**b**) the surface of the plate containing silver, which does not have electronic conductivity; and (**c**) the surface of the plate containing silver, which has electronic conductivity.

**Figure 5 polymers-15-03687-f005:**
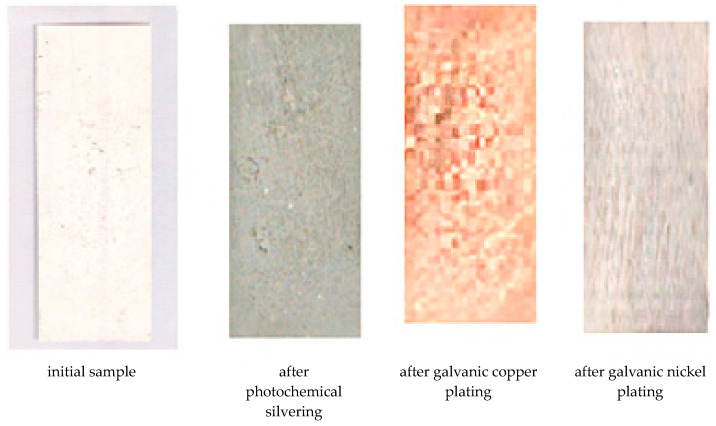
Photos of polymer samples after the operation of applying a silver film and subsequent electroplating with copper and nickel.

**Figure 6 polymers-15-03687-f006:**
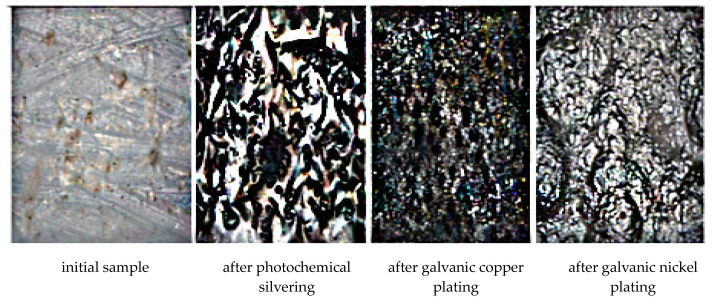
Images of the surface of polymer samples after the operation of applying a silver film and subsequent electroplating coatings with copper and nickel obtained by an electron microscope.

**Table 1 polymers-15-03687-t001:** Influence of different types of lighting on the duration of the formation of an electrically conductive silver layer.

No.	Type of Illumination of the Sample Surface, H	EMWLR Flow Density, W, W/m^2^	Silver Nitrate Concentration in the Sorption Layer, C_1_, g/L	Ascorbic Acid Concentration in the Sorption Layer, C_2_, g/L	Duration of the Process of Formation of an Electrically Conductive Silver Layer, τ, min
1	Darkness	0.001	10	10.05	475
2	Daylight	15.4	10	10.05	77
3	Light beam	725	10	10.05	21
4	Light beam	1080	10	10.05	13
5	Darkness	0.001	20	20.1	386
6	Daylight	10.8	20	20.1	25
7	Light beam	725	20	20.1	16
8	Light beam	1080	20	20.1	15

**Table 2 polymers-15-03687-t002:** Effect of the silver nitrate concentration in the surface sorption film on the formation of an electrically conductive silver layer.

No.	Volume of the Sorbed Surface of the Sample Solution, V, cm^3^	Initial Concentration of Silver Nitrate in the Sorption Layer, C_1_, g/L	Initial Concentration of Ascorbic acid in the Sorption Layer, C_2_, g/L	Thickness of the Electrically Conductive Silver Layer, ∆, microns	Silver Consumption for Obtaining an Electrically Conductive Layer, G, g/m^2^
1	145	5	5.025	-	-
2	145	8	8.03	-	-
3	145	10	10.05	0.11	1.1
4	150	20	20.1	0.23	2.3
5	150	30	30.15	0.35	3.5
6	150	40	40.1	0.48	4.8

**Table 3 polymers-15-03687-t003:** Quantity results.

Analytical	Result	[3-Sigma]	Met. calc.	Lin Int.(imp/s/µA)
Ag	82.276%	[0.251]	Quantitative-FP	AgKa 98.9225
Ti	16.548%	[0.164]	Quantitative-FP	TiKa 9.4935
Cu	0.827%	[0.025]	Quantitative-FP	CuKa 2.9098
Zn	0.349%	[0.021]	Quantitative-FP	ZnKa 1.4635

## Data Availability

The data used to support the findings of this study are included within the article.

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
