# Peer review of "Photochemical Metallization: Advancements in Polypropylene Surface Treatment"

_polymers, 2023, doi:10.3390/polym15183687_

Round 1
Reviewer 1 Report
Photochemical Metallization: Advancements in Polypropylene Surface Treatment" presents new methodologies for generating conductive coatings using various elements and techniques. This document provides adequate characterization. However, the document requires thorough checking of the English writing and chemical equations, as well as improvements to the images, as they appear to be of poor quality. In addition is important to check the technical vocabulary, there a lot of mistranslation. I have addressed some corrections and suggestions that I consider crucial in order to achieve an improved version of the manuscript.
Please, find in the text below some ingles corrections of wording and suggestions:
The research is devoted to the development of a technology for applying metal coatings to the surface of polypropylene products. At the same time, mechanical processing and etching were carried out according to methods known in the literature. The polymer was then treated with a solution of copper dichloride. As a result of this treatment, copper dichloride interacts with etching products to form associated active centers containing copper monochloride and partially reduced copper. After that, by dipping in a solution containing 10-20 g/l of silver nitrate and equivalent amounts of ascorbic acid (10.05-20.1 g/l), a sorption film of this solution was obtained on the polymer surface. A sample with a sorption film was exposed to electromagnetic waves of light radiation at a flux density of 700-1100 w/m2 and an electrically conductive film with a thickness of 0.11-0.48 microns was obtained for 15-20 minutes. Solar radiation and the radiation of electric lighting lamps were used as a source of electromagnetic waves. The suitability of electrically conductive films for electroplating and chemical metallization of the polymer surface was shown.
“The research is focused on developing a technology for applying metal coatings to the surface of polypropylene products. In this process, mechanical processing and etching were conducted following established methods in the literature. Subsequently, the polymer was treated with a solution of copper dichloride. During this treatment, copper dichloride reacts with the etching products, forming associated active centers that contain copper monochloride and partially reduced copper. Following this, the polymer sample was immersed in a solution containing 10-20 g/l of silver nitrate and equivalent amounts of ascorbic acid (10.05-20.1 g/l), resulting in a sorption film on the polymer surface. To achieve an electrically conductive film, the sample with the sorption film was exposed to electromagnetic waves of light radiation at a flux density of 700-1100 w/m2 for 15-20 minutes. Solar radiation and the radiation from electric lighting lamps were used as sources of electromagnetic waves. The study demonstrated that the electrically conductive films obtained were suitable for electroplating and chemical metallization of the polymer surface.”
Metal coatings are applied to the surfaces of polymer materials to give them new functional properties: electrical conductivity, hardness, wear resistance, etc. The methods of applying metal films to dielectric materials known in the literature are conventionally divided into physical and chemical.
Of the physical methods, thermal spraying is most often used. In this process, remolten metal particles are sprayed onto the coated surface with compressed air or gas. Liquid metal particles partially cool down on their way to the dielectric surface and only melt the surface by sticking to it [1-5].
A variation of this method is gas-phase, plasma and vacuum spraying [4,6]. The technology of metallization in vacuum is versatile, harmless to the environment and maintenance personnel. However, for its implementation, it is necessary to have rather complex equipment.
“Metal coatings are applied to polymer surfaces to enhance their functional properties, such as electrical conductivity, hardness, and wear resistance. These methods are commonly categorized as physical or chemical processes. Among physical methods, thermal spraying is frequently employed. This technique involves spraying molten metal particles onto the surface using compressed air or gas. The liquid metal particles partially cool during their trajectory and adhere to the dielectric surface, subsequently melting and forming a coating [1-5]. Gas-phase, plasma, and vacuum spraying are variations of this method [4,6]. Vacuum metallization technology is versatile, environmentally friendly, and safe for personnel. However, its implementation requires complex equipment.”
Physical methods also include magnetron sputtering in a vacuum chamber of metal particles (including silver) with subsequent application to the surface of the material. The method is based on the use of an anomalous glow discharge in an inert gas, in which positively charged ions formed in the discharge bombard the cathode surface in the erosion zone and knock out metal particles from it, which are then deposited as a thin layer on the surface of the processed material [7-9].
The metal coating is applied without the use of chemicals that pollute the environment. In addition, the high kinetic energy of the particles leaving the cathode surface ensures a good level of adhesion of the resulting film to the substrate. Disadvantage: the use of special expensive equipment. Thermal decomposition reactions can be used for metallization in the gas phase. The most suitable compounds for this purpose are metal carbonyls. During the reaction, undercertain conditions, they decompose, leaving metal on the coated surface and releasing carbon monoxide, which can again be used to produce metal carbonyl. That is, CO plays the role of a metal carrier reagent. This is not only convenient in terms of production, but also minimizes the unproductive costs of auxiliary reagents, eliminates environmental pollution. The wide spread of this method is hindered by the relative high temperature of the process and the high cost of metal carbonyls [10].
“Physical methods for metal coatings also include magnetron sputtering in a vacuum chamber with metal particles, including silver, followed by application to the material's surface. This technique utilizes an anomalous glow discharge in an inert gas, where positively charged ions generated in the discharge bombard the cathode surface in the erosion zone, ejecting metal particles. These particles then form a thin layer on the processed material's surface [7-9]. An advantage of this method is that it does not require the use of chemicals that could pollute the environment. Additionally, the high kinetic energy of the particles leaving the cathode surface ensures strong adhesion of the resulting film to the substrate. However, a drawback is the need for special and expensive equipment.
Thermal decomposition reactions can also be utilized for gas-phase metallization. Metal carbonyls are well-suited compounds for this purpose. Under certain conditions, they decompose, leaving metal on the coated surface and releasing carbon monoxide, which can be recycled to produce metal carbonyl again. Thus, CO acts as a metal carrier reagent. This not only simplifies production but also reduces the use of auxiliary reagents, minimizing environmental pollution. However, the wide adoption of this method is hindered by its relatively high process temperature and the cost of metal carbonyls [10].”
The main disadvantage of physical methods is the need to use complex expensive equipment. Therefore, the use of this method is economically justified when metallizing the same type of dielectric products in significant quantities (electronics, vehicles, etc.).
Chemical methods of metallization of dielectrics consist in carrying out chemical processes for obtaining metal particles in a gas or liquid medium.
In [11, 12], copper phosphide films were obtained on fabric materials moistened with copper sulfate by exposure to gaseous phosphine, which were then transformed into silver. The possibility of using this method for both synthetic and natural fabrics is shown. The disadvantage is the use of toxic phosphine as a reducing agent and the resulting need to work in sealed conditions.
“The main drawback of physical methods is the requirement for complex and expensive equipment. Therefore, the economical justification for using these methods arises when metallizing large quantities of the same type of dielectric products, such as in electronics or vehicles. On the other hand, chemical methods for metallizing dielectrics involve carrying out chemical processes to obtain metal particles in a gas or liquid medium.
In [11, 12], copper phosphide films were obtained on fabric materials by treating them with copper sulfate and then exposing them to gaseous phosphine, which subsequently transformed into silver. The potential for using this method with both synthetic and natural fabrics is demonstrated. However, a disadvantage lies in the use of toxic phosphine as a reducing agent, necessitating work in sealed conditions to ensure safety.”
Line 60 … into silver… that sentence is incorrect or mistranslation, please
When you talk about a references as this “In [11, 12],…” please make a little descripction about the reference followed by the this. Example the author name…
Please correct the chemical reactions with the adequate nomenclature and way to write them
Please correct of the symbol of light (hv, γ)
How did you determinate the reduction of Cu(2+) to Cu(1+)?
What did you mean with “The vacancies (+ ), line 179”?
Please check the chemical reactions, they look incorrect
In line 191 ”SP” change into “Ksp”
Line 198 “As a result of these photochemical reactions, silver is formed” maybe you mean “as a result of these photochemical reactions, silver ions are reduced to metallic silver”
Please improve the quality of the graphs and images
no comments
Author Response
REVIEWER 1
Comments and Suggestions for Authors
Photochemical Metallization: Advancements in Polypropylene Surface Treatment" presents new methodologies for generating conductive coatings using various elements and techniques. This document provides adequate characterization. However, the document requires thorough checking of the English writing and chemical equations, as well as improvements to the images, as they appear to be of poor quality. In addition is important to check the technical vocabulary, there a lot of mistranslation. I have addressed some corrections and suggestions that I consider crucial in order to achieve an improved version of the manuscript.
Please, find in the text below some ingles corrections of wording and suggestions:
The research is devoted to the development of a technology for applying metal coatings to the surface of polypropylene products. At the same time, mechanical processing and etching were carried out according to methods known in the literature. The polymer was then treated with a solution of copper dichloride. As a result of this treatment, copper dichloride interacts with etching products to form associated active centers containing copper monochloride and partially reduced copper. After that, by dipping in a solution containing 10-20 g/l of silver nitrate and equivalent amounts of ascorbic acid (10.05-20.1 g/l), a sorption film of this solution was obtained on the polymer surface. A sample with a sorption film was exposed to electromagnetic waves of light radiation at a flux density of 700-1100 w/m2 and an electrically conductive film with a thickness of 0.11-0.48 microns was obtained for 15-20 minutes. Solar radiation and the radiation of electric lighting lamps were used as a source of electromagnetic waves. The suitability of electrically conductive films for electroplating and chemical metallization of the polymer surface was shown.
“The research is focused on developing a technology for applying metal coatings to the surface of polypropylene products. In this process, mechanical processing and etching were conducted following established methods in the literature. Subsequently, the polymer was treated with a solution of copper dichloride. During this treatment, copper dichloride reacts with the etching products, forming associated active centers that contain copper monochloride and partially reduced copper. Following this, the polymer sample was immersed in a solution containing 10-20 g/l of silver nitrate and equivalent amounts of ascorbic acid (10.05-20.1 g/l), resulting in a sorption film on the polymer surface. To achieve an electrically conductive film, the sample with the sorption film was exposed to electromagnetic waves of light radiation at a flux density of 700-1100 w/m2 for 15-20 minutes. Solar radiation and the radiation from electric lighting lamps were used as sources of electromagnetic waves. The study demonstrated that the electrically conductive films obtained were suitable for electroplating and chemical metallization of the polymer surface.”
Metal coatings are applied to the surfaces of polymer materials to give them new functional properties: electrical conductivity, hardness, wear resistance, etc. The methods of applying metal films to dielectric materials known in the literature are conventionally divided into physical and chemical.
Of the physical methods, thermal spraying is most often used. In this process, remolten metal particles are sprayed onto the coated surface with compressed air or gas. Liquid metal particles partially cool down on their way to the dielectric surface and only melt the surface by sticking to it [1-5].
A variation of this method is gas-phase, plasma and vacuum spraying [4,6]. The technology of metallization in vacuum is versatile, harmless to the environment and maintenance personnel. However, for its implementation, it is necessary to have rather complex equipment.
“Metal coatings are applied to polymer surfaces to enhance their functional properties, such as electrical conductivity, hardness, and wear resistance. These methods are commonly categorized as physical or chemical processes. Among physical methods, thermal spraying is frequently employed. This technique involves spraying molten metal particles onto the surface using compressed air or gas. The liquid metal particles partially cool during their trajectory and adhere to the dielectric surface, subsequently melting and forming a coating [1-5]. Gas-phase, plasma, and vacuum spraying are variations of this method [4,6]. Vacuum metallization technology is versatile, environmentally friendly, and safe for personnel. However, its implementation requires complex equipment.”
Physical methods also include magnetron sputtering in a vacuum chamber of metal particles (including silver) with subsequent application to the surface of the material. The method is based on the use of an anomalous glow discharge in an inert gas, in which positively charged ions formed in the discharge bombard the cathode surface in the erosion zone and knock out metal particles from it, which are then deposited as a thin layer on the surface of the processed material [7-9].
The metal coating is applied without the use of chemicals that pollute the environment. In addition, the high kinetic energy of the particles leaving the cathode surface ensures a good level of adhesion of the resulting film to the substrate. Disadvantage: the use of special expensive equipment. Thermal decomposition reactions can be used for metallization in the gas phase. The most suitable compounds for this purpose are metal carbonyls. During the reaction, undercertain conditions, they decompose, leaving metal on the coated surface and releasing carbon monoxide, which can again be used to produce metal carbonyl. That is, CO plays the role of a metal carrier reagent. This is not only convenient in terms of production, but also minimizes the unproductive costs of auxiliary reagents, eliminates environmental pollution. The wide spread of this method is hindered by the relative high temperature of the process and the high cost of metal carbonyls [10].
“Physical methods for metal coatings also include magnetron sputtering in a vacuum chamber with metal particles, including silver, followed by application to the material's surface. This technique utilizes an anomalous glow discharge in an inert gas, where positively charged ions generated in the discharge bombard the cathode surface in the erosion zone, ejecting metal particles. These particles then form a thin layer on the processed material's surface [7-9]. An advantage of this method is that it does not require the use of chemicals that could pollute the environment. Additionally, the high kinetic energy of the particles leaving the cathode surface ensures strong adhesion of the resulting film to the substrate. However, a drawback is the need for special and expensive equipment.
Thermal decomposition reactions can also be utilized for gas-phase metallization. Metal carbonyls are well-suited compounds for this purpose. Under certain conditions, they decompose, leaving metal on the coated surface and releasing carbon monoxide, which can be recycled to produce metal carbonyl again. Thus, CO acts as a metal carrier reagent. This not only simplifies production but also reduces the use of auxiliary reagents, minimizing environmental pollution. However, the wide adoption of this method is hindered by its relatively high process temperature and the cost of metal carbonyls [10].”
The main disadvantage of physical methods is the need to use complex expensive equipment. Therefore, the use of this method is economically justified when metallizing the same type of dielectric products in significant quantities (electronics, vehicles, etc.). Chemical methods of metallization of dielectrics consist in carrying out chemical processes for obtaining metal particles in a gas or liquid medium.
In [11, 12], copper phosphide films were obtained on fabric materials moistened with copper sulfate by exposure to gaseous phosphine, which were then transformed into silver. The possibility of using this method for both synthetic and natural fabrics is shown. The disadvantage is the use of toxic phosphine as a reducing agent and the resulting need to work in sealed conditions.
“The main drawback of physical methods is the requirement for complex and expensive equipment. Therefore, the economical justification for using these methods arises when metallizing large quantities of the same type of dielectric products, such as in electronics or vehicles. On the other hand, chemical methods for metallizing dielectrics involve carrying out chemical processes to obtain metal particles in a gas or liquid medium.
In [11, 12], copper phosphide films were obtained on fabric materials by treating them with copper sulfate and then exposing them to gaseous phosphine, which subsequently transformed into silver. The potential for using this method with both synthetic and natural fabrics is demonstrated. However, a disadvantage lies in the use of toxic phosphine as a reducing agent, necessitating work in sealed conditions to ensure safety.”
ANSWER: Appropriate changes have been made to the text of the manuscript. Highlighted in red.
Line 60 … into silver… that sentence is incorrect or mistranslation, please
ANSWER: Appropriate changes have been made to the text of the manuscript. Highlighted in red.
When you talk about a references as this “In [11, 12],…” please make a little descripction about the reference followed by the this. Example the author name…
ANSWER: Appropriate changes have been made to the text of the manuscript. Highlighted in red.
Please correct the chemical reactions with the adequate nomenclature and way to write them
ANSWER: All chemical reactions in the revised version of the article are written in accordance with the nomenclature.
Please correct of the symbol of light (hv, γ)
ANSWER: Corrections have been made in reactions 2 and 5. Symbols v have been replaced by hv.
How did you determinate the reduction of Cu(2+) to Cu(1+)?
ANSWER: The mechanism for the reduction of Cu(2+) to Cu(1+) is shown in our previous work.
M.S. Sataev, Sh.T. Koshkarbaeva, K.B. Amanbaeva, P.A. Abdurazova , G.T.Assilbekova, Y.B. Raiymbekov and D.Urazkeldieva. Photochemical reduction of copper and silver ions on cellulose-containing fabrics Rasayan J. Chem., 15(1), 31-40(2022) http://dx.doi.org/10.31788/RJC.2022.1516595
What did you mean with “The vacancies (+ ), line 179”?
ANSWER: By vacancies (+) we meant electron holes in semiconductors. This has been eliminated and in the corrected version the word vacancy has been replaced by the word electron hole.
Please check the chemical reactions, they look incorrect
ANSWER: All chemical reactions in the revised version of the article are written in accordance with the nomenclature.
In line 191 ”SP” change into “Ksp”
ANSWER: Appropriate changes have been made to the text of the manuscript. Highlighted in red.
Line 198 “As a result of these photochemical reactions, silver is formed” maybe you mean “as a result of these photochemical reactions, silver ions are reduced to metallic silver”
ANSWER: Appropriate changes have been made to the text of the manuscript. Highlighted in red.
Please improve the quality of the graphs and images.
ANSWER: The graphs and images have been replaced taking into account the comments.
Reviewer 2 Report
In order to enhance the quality of the presented manuscript, authors are advised to address following issues. The Abstract section (lines 11-17) is written as a part of Experimental section. There is no need to describe in details the procedure of treatment of the PP in the Abstract; please rewrite the Abstract section! In the Introduction section authors are citing the literature as “In [11, 12], copper phosphide films…”. It is better to use phrases like “In the literature”, or “In the work of Name…” At the end of this section, lines 115-118, authors highlighted the main goals of this work. However, they are starting it as: “The given review shows…” Is this a review? In the Materials and Methods section authors are mentioning the thickness of the sorption film (400-450 microns) and of conductive silver film (0.1-0.48 microns); lines 130-134. However, there are no details how they determined the mentioned thickness. Page 4, lines 149-158, more details about SEM, EDX and testing methods referred in the [24] and [25] are needed. Page 5, lines 216-217, authors repeated twice the same phrase: “that the reduction of silver”. Figs 2-5 are not clear, resolution too low; please enhance it. On the Fig. 2 there are some words on the Russian; please translate it to the English. Likewise, authors focused (by red arrow) on the specific area to highlight the composition. Can you magnify that area? After all, this is SEM. What about accumulations or agglomerations in the upper right corner of the Fig. 2? In order to better understand the changes, authors are advised to perform SEM analysis of the surface of PP before treatment; as they did on the Fig 3 and 5. The photos showed on the Fig. 5 are taken by camera? Did you use optical microscope? They are really unclear. It is important to show the changes that you are trying to present in this work. These Figs are not convincing enough. You need to verify the distribution and thickness of the sorption layers on the polymer surface. Perhaps to use another technique beside SEM? By XRD authors only verified the composition, there is no info about distribution or homogeneity of films. Finally, page 9, line 310, authors are concluding that: “Such a film with a thickness of 0.11-0.48 microns is suitable for further expansion of the metal layer.” Once again, how did you determined the abovementioned thickness?
Moderate editing.
Author Response
REVIEWER 2
Comments and Suggestions for Authors
In order to enhance the quality of the presented manuscript, authors are advised to address following issues. The Abstract section (lines 11-17) is written as a part of Experimental section. There is no need to describe in details the procedure of treatment of the PP in the Abstract; please rewrite the Abstract section!
ANSWER: Appropriate changes have been made to the text of the manuscript. Highlighted in red.
In the Introduction section authors are citing the literature as “In [11, 12], copper phosphide films…”. It is better to use phrases like “In the literature”, or “In the work of Name…”
ANSWER: Appropriate changes have been made to the text of the manuscript. Highlighted in red.
At the end of this section, lines 115-118, authors highlighted the main goals of this work. However, they are starting it as: “The given review shows…” Is this a review?
ANSWER: The word "review" indicates a review of the literature in the introduction regarding various methods for the metallization of dielectrics.
In the Materials and Methods section authors are mentioning the thickness of the sorption film (400-450 microns) and of conductive silver film (0.1-0.48 microns); lines 130-134. However, there are no details how they determined the mentioned thickness.
ANSWER: Appropriate changes have been made to the text of the manuscript. Highlighted in red.
To determine the thickness of the sorption film, a plate with a sizer of 10 × 10 cm was cut out of the polymer and this plate was lowered into the vessel with the test solution, then the sample was taken out and slightly shaken over the vessel. At the same time, the volume of the solution decreased. In order for the reduction to be measurable, these operations were repeated more than 10 times. The one-time carryover of the solution determined in this way by this plate was 8-9 cm3. The division of this value by the area of the plate 10∙10∙2=200 cm2 gives us the thickness of the sorption film 0.04-0.045 cm or 400-450 µm.
When determining the thickness of the silver film, the weight method used in electroplating was used. To do this, on an analytical balance, the weight of the sample was lo and after the experiment. The resulting weight gain was divided by the product of the area of the sample and the density of silver.
Page 4, lines 149-158, more details about SEM, EDX and testing methods referred in the [24] and [25] are needed.
ANSWER: Appropriate changes have been made to the text of the manuscript. Highlighted in red.
The samples were examined under an ISM-6490-LV Scanning Electron Microscope. This instrument facilitates high-resolution imaging of specimen surfaces using a focused electron beam.
Elemental analysis was performed using an EDX-7000 Energy Dispersive X-ray Spectroscopy system manufactured by Shimadzu Corporation, Japan. The specimen was placed within the EDX instrument. Upon irradiation by an electron beam, the sample emitted characteristic X-rays. The EDX-7000 system detected and analyzed these X-rays to determine their energy and count rates.
The electrical resistance of the films was measured using a DT-830B digital multimeter, a versatile tool capable of assessing resistance, voltage, and current. To measure resistance, the DT-830B was set to the appropriate resistance scale (ohmmeter mode). The probes of the multimeter were then contacted with different parts of the film, ensuring a stable and firm connection.
The resulting conductive films were tested for the processes of galvanic copper plating, nickel plating, as well as in the process of chemical nickel plating [24]. Conductive films were first prepared by immersing them in an electrolyte bath containing copper ions, typically sourced from copper sulfate. Following thorough cleansing to ensure the removal of contaminants, the films acted as the cathode in an electroplating system. An electric current was then introduced, resulting in the deposition of a copper layer onto the conductive film surface. To evaluate the quality of adhesion between the electroplated coatings and the polymer base, a specific test as detailed in [25] was employed.
Page 5, lines 216-217, authors repeated twice the same phrase: “that the reduction of silver”.
ANSWER: Appropriate changes have been made to the text of the manuscript. Highlighted in red.
Figs 2-5 are not clear, resolution too low; please enhance it.
ANSWER: The graphs and images have been replaced taking into account the comments.
On the Fig. 2 there are some words on the Russian; please translate it to the English. Likewise, authors focused (by red arrow) on the specific area to highlight the composition. Can you magnify that area? After all, this is SEM. What about accumulations or agglomerations in the upper right corner of the Fig. 2? In order to better understand the changes, authors are advised to perform SEM analysis of the surface of PP before treatment; as they did on the Fig 3 and 5. The photos showed on the Fig. 5 are taken by camera? Did you use optical microscope? They are really unclear. It is important to show the changes that you are trying to present in this work. These Figs are not convincing enough. You need to verify the distribution and thickness of the sorption layers on the polymer surface. Perhaps to use another technique beside SEM?
ANSWER: Comments have been taken into account. The picture has been replaced with a better one.
By XRD authors only verified the composition, there is no info about distribution or homogeneity of films.
ANSWER: The formation of an electrically conductive film of metallic silver is also confirmed by X-ray phase analysis. Figure 3 and Table 3 shows that, in addition to the silver peak, there appears a peak of copper, the compound of which was used for activation. The rest of the peaks are related to the elements that were introduced into the polymer composition during its preparation.
Finally, page 9, line 310, authors are concluding that: “Such a film with a thickness of 0.11-0.48 microns is suitable for further expansion of the metal layer.” Once again, how did you determined the abovementioned thickness?
ANSWER: In the Materials and Methods section, the thickness determination method has been described in detail.
Round 2
Reviewer 2 Report
Dear Authors, thank you for considering my comments and suggestions.
Moderate editing.